# DEEP BANACH SPACE KERNELS

## ABSTRACT

The recent success of deep learning has encouraged many researchers to explore the deep/concatenated variants of classical kernel methods. Some of which includes MLMKL, DGP and DKL. These methods have proven to be quite helpful in various real-world settings. But so far, we have only been utilizing kernels from Hilbert spaces, which has their own limitations. In this paper, we address these shortcomings by introducing a new concatenated kernel learning approach that uses the kernels from the reproducing kernel Banach spaces(RKBSs) instead. We present a general framework of construction for these Deep RKBS models and then provide a representer theorem for regularized learning problems. We also describe the relationship with its deep RKHS variant as well as standard deep Gaussian processes. In the end, we construct and implement a two-layer deep RKBS model and demonstrate it on a range of machine learning tasks.

## 1 INTRODUCTION

In recent years, promising new variants of kernel learning methods, namely deep ker-nel learning and multi-layer-MKL (MLMKL) algorithms have been developed. These concatenated kernel learning approach includes, see e.g. Cho & Saul (2009); Damianou & Lawrence (2013); Zhuang et al. (2011). Although they have proven to be very successful in regression and classification tasks. we are still only utilizing kernel from Hilbert Space. There has be a recent interest in studying a space which is similar to RKHS but contains alot more functions i.e, reproducing kernel Banach space (RKBS). In this paper, we will define a general framework of concatenated RKBSs as well as provide a concatenated representer theorem which can be used to develop a generalized variant of all the concatenated kernel learning(Bohn et al. (2019)) methods described earlier. We will now give a brief overview of our paper.

In Section 2, we briefly review the interpolation problem in RKBS and discuss how we can recast this to RKHS. In Section 3, we introduce the optimal concatenated approximation problem for arbitrary loss function and regularizers. We first present a general framework to construct a deep/concatenated RKBS and then propose a representer theorem for this problem in multi-layer case. we then examine the concrete example of a two-layer RKBS kernel. Furthermore, we discuss the relation of our method to classical concatenated kernel learning in RKHSs which describe the connect with multiple array of methods such as DeepGP, DKL, MLMKL etc. In section 4 we look at a simple experiment and compare our Deep RKBS kernels with Deep RKHS ones. we also briefly talk about our library we have built to easily perform experimentation with Deep RKBS Kernels. Finally, we conclude with some related works and a summary of our paper.

## 2 REPRODUCING KERNEL BANACH SPACES

In this section, we briefly define RKBS and its reproducing kernel then we will present the representer theorem for minimal norm interpolation problem. for more comprehensive overview we refer the readers to Lin et al. (2019b).

A *reproducing kernel Banach space* $\mathcal{B}$ on a prescribed nonempty set $X$ is a Banach space of certain functions on $X$ such that every point evaluation functional $\delta_x$, $x \in X$ on $\mathcal{B}$ is continuous, that is, there exists a positive constant $C_x$ such that

$$|\delta_x(f)| = |f(x)| \leq C_x \|f\|_{\mathcal{B}} \text{ for all } f \in \mathcal{B}.$$

this definition is the natural generalization of the classical definition of RKHS. Now before moving forward, we will present another definition as well as general framework of construction for RKBS which we will be using throughout this paper.

(**Construction of RKBS using feature maps**) Let $\mathcal{W}^{(1)}, \mathcal{W}^{(2)}$ be two Banach spaces, and $\langle \cdot, \cdot \rangle_{\mathcal{W}^{(1)} \times \mathcal{W}^{(2)}}$ be a continuous bilinear form on $\mathcal{W}^{(1)} \times \mathcal{W}^{(2)}$. Suppose there exist two nonempty sets $\Omega^{(1)}$ and $\Omega^{(2)}$, and mappings $\Phi^{(1)} : \Omega^{(1)} \to \mathcal{W}^{(1)}, \quad \Phi^{(2)} : \Omega^{(2)} \to \mathcal{W}^{(2)}$ such that with respect to the bilinear form $\operatorname{span} \Phi^{(1)}(\Omega^{(1)})$ is dense in $\mathcal{W}^{(1)}$, $\operatorname{span} \Phi^{(2)}(\Omega^{(2)})$ is dense in $\mathcal{W}^{(2)}$. We construct

$$\mathcal{B}^{(1)} := \left\{ f_v(x) := \langle \Phi^{(1)}(x), v \rangle_{\mathcal{W}^{(1)} \times \mathcal{W}^{(2)}} : v \in \mathcal{W}^{(2)}, x \in \Omega^{(1)} \right\} \tag{2.1}$$

and

$$\mathcal{B}^{(2)} := \left\{ g_u(y) := \langle u, \Phi^{(2)}(y) \rangle_{\mathcal{W}^{(1)} \times \mathcal{W}^{(2)}} : u \in \mathcal{W}^{(1)}, y \in \Omega^{(2)} \right\} \tag{2.2}$$

with norm $\|f_v\|_{\mathcal{B}^{(1)}} := \|v\|_{\mathcal{W}^{(2)}}$ and $\|g_u\|_{\mathcal{B}^{(2)}} := \|u\|_{\mathcal{W}^{(1)}}$ respectively.

**Theorem 2.1** *Let $\mathcal{B}^{(1)}$ and $\mathcal{B}^{(2)}$ be constructed as above. Then with the bilinear form on $\mathcal{B}^{(1)} \times \mathcal{B}^{(2)}$*

$$\langle f_v, g_u \rangle_{\mathcal{B}^{(1)} \times \mathcal{B}^{(2)}} := \langle u, v \rangle_{\mathcal{W}^{(1)} \times \mathcal{W}^{(2)}} \text{ for all } f_v \in \mathcal{B}^{(1)} \text{ and all } g_u \in \mathcal{B}^{(2)}, \tag{2.3}$$

*$\mathcal{B}^{(1)}$ is an RKBS on $\Omega^{(1)}$ with the adjoint RKBS $\mathcal{B}^{(2)}$ on $\Omega^{(2)}$. Moreover,*

$$K(x,y) := \langle \Phi^{(1)}(x), \Phi^{(2)}(y) \rangle_{\mathcal{W}^{(1)} \times \mathcal{W}^{(2)}}, \ x \in \Omega^{(1)}, y \in \Omega^{(2)}, \tag{2.4}$$

*is a reproducing kernel for $\mathcal{B}^{(1)}$. Using the reproducing property we can rewrite $f \in \mathcal{B}^{(1)}$ as:*

$$f(x) = \langle f, K(x, \cdot) \rangle_{\mathcal{B}_1 \times \mathcal{B}_2} \text{ for all } x \in \Omega^{(1)} \text{ and all } f \in \mathcal{B}_1, \tag{2.5}$$

*where, $K(x, \cdot) \in \mathcal{B}^{(2)}$.*

## 2.1 Interpolation Problem in RKBS

The minimal norm interpolation problem looks for the minimizer

$$f_{\text{inf}} := \arg \inf_{f \in S_{\mathbf{x}, \mathbf{t}}} \|f\|_{\mathcal{B}^{(1)}} \text{ where } S_{\mathbf{x}, \mathbf{t}} = \left\{ f \in \mathcal{B}^{(1)} : f(x_j) = t_j, \ j \in \mathbb{N}_m \right\} \tag{2.6}$$

$$v_{\text{inf}} := \arg \inf_{v \in V_{\mathbf{x}, \mathbf{t}}} \|v\|_{\mathcal{W}^{(2)}} \tag{2.7}$$

with

$$V_{\mathbf{x}, \mathbf{t}} := \left\{ v \in \mathcal{W}^{(2)} : \langle \Phi^{(1)}(x_j), v \rangle_{\mathcal{W}^{(1)} \times \mathcal{W}^{(2)}} = t_j, \ j \in \mathbb{N}_m \right\}. \tag{2.8}$$

**Theorem 2.2** *(**Representer Theorem**) Assume the same assumptions as in Theorem 2.1. In addition, suppose that $\mathcal{W}^{(2)}$ is reflexive, strictly convex and Gâteaux differentiable, and the set $\{\Phi^{(1)}(x_j) : j \in \mathbb{N}_m\}$ is linearly independent in $\mathcal{W}^{(1)}$. Then the minimal norm interpolation problem (2.7) has a unique solution $v_{\text{inf}} \in \mathcal{W}^{(2)}$ and it satisfies*

$$\mathcal{G}(v_{\text{inf}}) \in \left( (\Phi^{(1)}(\mathbf{x}))^{\vdash} \right)^{\perp}. \tag{2.9}$$

If $\Omega^{(1)} = \Omega^{(2)}$, $\mathcal{W}^{(2)*} = \mathcal{W}^{(1)}$ and $\mathcal{W}^{(2)}$ is reduced to a Hilbert Space then we can recover the classical representer theorem(Schölkopf et al. (2002)) for minimal norm interpolation in RKHS.

## 3 A representer Theorem for concatenated kernel Learning in Banach Space

In this section, we will be deriving a *concatenated* RKBS representer theorem for an arbitrary number $L \in \mathbb{N}$ of concatenations of *vector-valued* RKBS spaces. For more comprehensive treatment of vector-valued RKBS we refer the readers to Chen et al. (2019), Zhang & Zhang (2013) and Lin et al. (2019a) respectively.

Let $B_1, ..., B_L$ be reproducing kernel Banach Spaces with finite dimensional domain $\Omega_l^{(1)}$ and ranges $R_l \subseteq \mathbb{R}^{d_l}$ with $d_l \in \mathbb{N}$ for $l = 1, ..., L$ such that $R_l \subseteq \Omega_{l-1}^{(1)}$ for $l = 2, ..., L, \Omega_L^{(1)} = \Omega$ and $R_1 \subseteq \mathbb{C}$. We consider learning a function from a prescribed set of finite sampling data

$$\mathbf{z} := \{(x_i, t_i) : i \in N\} \subseteq \Omega \times \mathbb{C}$$

Let furthermore $\mathcal{L} : \mathbb{R}^2 \to [0, \infty]$ be an arbitrary continuous and convex loss function and let $\Theta_1, ..., \Theta_L : [0, \infty)$ be continuous, convex and strictly monotonically increasing functions. For each arbitrary function $f_l \in \mathcal{B}_l : \forall l = 1, ...L$, we set

$$J(f_1, ..., f_L) := \sum_{i=1}^{N} \mathcal{L}(t_i, f_1 \circ ... \circ f_L(x_i)) + \sum_{l=1}^{L} \Theta_l(\|f_l\|_{B_l}) \tag{3.10}$$

and our objective is:

$$\inf_{f_1 \circ ... \circ f_L \in B_1 \times ... \times B_L} J(f_1 \circ ... \circ f_L) \tag{3.11}$$

Since, $\forall l = 1, ..., L$

$$f_l(x^{(l)}) := f_{v_l}(x^{(l)}) = \langle \Omega_l^{(1)}(x^{(l)}), v_l \rangle_{W_l^{(1)} \times W_l^{(2)}}, x^{(l)} \in \Omega_l^{(1)}, v_l \in \bar{\text{span}}\{\Phi_l^{(2)}(y^{(l)}) : y^{(l)} \in \Omega_l^{(2)}\}$$

Thus 3.11 reduces to

$$v_1, ...v_{L\inf} := \arg\inf_{v_1, ..., v_L \in W_1^{(2)}, ..., W_L^{(2)}} \mathcal{L}\left(t_i, \langle \Phi_1^{(1)}, v_1 \rangle_{W_1^{(1)} \times W_1^{(2)}} \circ ... \circ \langle \Phi_L^{(1)}(x_i), v_L \rangle_{W_L^{(1)} \times W_L^{(2)}}\right)$$
$$+ \sum_{l=1}^{L} \Theta_l(\|v_l\|_{W_l^{(2)}}) \tag{3.12}$$

Even if $\mathcal{L}$ is a convex loss function (3.11) is still a highly non-linear optimization problem. we therefore assume that, there are $w \in \mathbb{N}$ optimal composite functions which minimizes $J$. We Let $F_l^* = f_1^{l*}, ..., f_w^{l*}$ be a set of all the optimal functions in $B_l$ for all $l = 1, ..., L$ .

**Theorem 3.1** *In addition to above assumption, suppose that $W_l^{(2)}$ is reflexive, strictly convex and Gâteaux differentiable for all $l = 1, ...L$, and the set $\Phi_l^{(1)}(x_j) : x_j \in D_l, \forall j \in N$ is linearly independent in $W_l^{(1)}$ . Provided that $F_l^*$ is non-empty for all $l = 1, ..., L$ then, there exist a set of $w$ minimizers where each $v_{l\inf} \in W_l^{(2)} : \forall l = 1, ...L$ satisfies*

$$\mathcal{G}(v_{l\inf}) \in (V_{lx,0})^{\perp}$$

*where,*

$$V_{lx,0} := \{v_l \in W_l^{(2)} : \langle \Phi_l^{(1)}(x_j^{(l)}), v_l \rangle_{W_l^{(1)} \times W_l^{(2)}} = 0 : x_j^{(l)} \in \Omega_l^{(1)}, j \in N\}$$

*Proof:* Suppose there exist a minimizer $f_l^* \in F_l^* \in B_l$ then we create a data set $D_l$:
$$D_l := (x_j, f_l^*(x_j)) : j \in \mathbb{N}_m$$

By theorem 2.2 there exists a unique solution $v_{inf} \in W_2$ for the min norm interpolation problem with the samples $D_l$ and it satisfy that $\mathcal{G}(v_{l\inf}) \in (V_{lx,0})^{\perp}$ It follows that $f_{lv\inf} = \langle \Phi_l^{(1)}(\cdot), v_{l\inf} \rangle_{W_l^{(1)} \times W_l^{(2)}}$ interpolates the sample data $D_l$ and for all $v_l \in W_l^{(2)}$:

$$||v_{linf}||_{W_l^{(2)}} \leq ||v_l||_{W_l^{(2)}}$$

Thus, $f_{v\inf}(x) = f_{v^*}(x)$.

We can extend this result for all $l = 1, ..., L$ such that:

$$\mathcal{G}(\hat{v}_{l\text{inf}}) \in (V_{l_{x,0}})^{\perp}$$

as long as $F_l^* \in B_l$ is non-empty. The proof is complete. $\square$ $\hfill\square$

## 3.1 EXAMPLE

In this section, we will define a concrete example of a 2 layer RKBS using our framework of construction. Suppose, our inner vector-valued RKBS $B_2$ is endowed by $l_1$ norm and our outer vector-valued RKBS $B_1$ is endowed by $L_p$ norm where, $1 < p < \infty$. using the representer theorem (see, Lin et al. (2019a) and Chen et al. (2019) ) we can rewrite a vector-valued $f_2(\cdot) \in B_2$ as:

$$f_2(\cdot) = \sum_{i=1}^{N} \sum_{k_2=1}^{d_2} c_{i,k_2} K_2(\boldsymbol{x}_i, \cdot) \boldsymbol{e}_{k_2}$$

for certain coefficients $c_{i,k_2} \in \mathbb{R}$. Furthermore, we have that $f_1 \in \mathcal{S}_1^{\mathbf{x}} = \text{span}\{K_1(f_2(\boldsymbol{x}_i), \cdot) \mid i = 1, \ldots, N\}$ and thus

$$f_1(\cdot) = \sum_{j=1}^{N} \alpha_j K_1 \left( \sum_{i=1}^{N} \sum_{k_2=1}^{d_2} c_{i,k_2} K_2(\boldsymbol{x}_i, \boldsymbol{x}_j) \boldsymbol{e}_{k_2}, \cdot \right)$$

The concatenated function is then given by $h(\cdot) := f_1 \circ f_2(\cdot) = \sum_{j=1}^{N} \alpha_j \mathcal{K}(\boldsymbol{x}_j, \cdot)$ with the following **deep RKBS kernel**

$$\mathcal{K}(\boldsymbol{x}, \boldsymbol{y}) = K_1 \left( \sum_{i=1}^{N} \sum_{k_2=1}^{d_2} c_{i,k_2} K_2(\boldsymbol{x}_i, \boldsymbol{x}) \boldsymbol{e}_{k_2}, \sum_{i=1}^{N} \sum_{k_2=1}^{d_2} c_{i,k_2} K_2(\boldsymbol{x}_i, \boldsymbol{y}) \boldsymbol{e}_{k_2} \right) \quad (3.13)$$

Therefore, instead of considering the infinite-dimensional optimization problem of finding $f_1 \in B_1$ and $f_2 \in B_2$ that minimize

$$J(f_1, f_2) = \sum_{i=1}^{N} \mathcal{L}(y_i, f_1(f_2(\boldsymbol{x}_i))) + \Theta_1 \left( \|f_1\|_{B_1}^2 \right) + \Theta_2 \left( \|f_2\|_{B_2}^2 \right)$$

we can restrict ourselves to finding the $N + N \cdot d_2$ coefficients $\alpha_j, c_{i,k_2}$ for $i, j = 1, \ldots, N$ and $k_2 = 1, \ldots, d_2$

## 3.2 RELATION TO DEEP KERNEL LEARNING

As briefly mentioned in 2.2, we can recover the classical representer theorem for interpolation problem ( the detailed proof is given in the appendix ). If we assume $B_1, ...B_L$ to be $\mathcal{H}_1, ..., \mathcal{H}_L$ where $\mathcal{H}_l \ \forall l = 1, ...L$ as a set of RKHS of vector-valued function then we can find a set of minimizers $f_1, ..., f_L \ f_l \in \mathcal{H}_l$ of 3.10 which satisfies that for all $l = 1, ...L$:

$$f_l \in \text{span } K_l(f_{l+1} \circ ... \circ f_L(x_i), \cdot) e_{kl} : i = 1, ...N \text{and} k_l = 1, ..., d_l \quad (3.14)$$

where $K_l$ denotes the reproducing kernel of $\mathcal{H}_l$ and $e_{kl} \in \mathbf{R}^{d_l}$ is the $k_l$-th unit vector.

which is exactly the representer theorem for concatenated kernel learning in hilbert space as described in Bohn et al. (2019), if we define a probability measure on each $\mathcal{H}_l$ above then we can recover the Deep gaussian processes. which means that the methods such as Cho & Saul (2009), Damianou & Lawrence (2013), Zhuang et al. (2011), Wilson et al. (2016) are the special cases of our framework. It will be interesting to see the RKBS equivalent of these methods using our framework in future.

## 4 EXPERIMENT

Although, we are leaving more thorough experimentation on deep RKBS kernels for future works. Here, for the sake of completeness we simply perform some preliminary experiments on two synthetic data which is generated as follows:

We choose $\Omega = [0, 1]^2$

$$h_1 : \Omega \to \mathbb{R} \qquad h_1(x, y) := (0.1 + |x - y|)^{-1}$$

$$h_2 : \Omega \to \mathbb{R} \qquad h_2(x, y) := \begin{cases} 1 & \text{if } x \cdot y > \frac{3}{20} \\ 0 & \text{else} \end{cases}$$

We define our deep RKBS kernel similar to 3.13 where we use the following kernels:

$$K_2(x, y) := 1 - |x - y|$$

which is the reproducing kernel for C([0,1]) and for $K_1$:

$$K_1(x, y) := \min\{x, y\} - t * xy : 0 < t < 1$$

which is the generalization of Brownian bridge kernel. now, for deep RKHS we take the polynomial kernel of degree 2 as inner kernel and Matern kernel as its outer kernel.

We compare deep RKHS and deep RKBS in regularized regression setting which is regularized by $l^2$. the results are presented in table below where the performance is calculated by averaging the MSE loss of 50 trials.

|  | avg. MSE |
| --- | --- |
| Deep RKHS | 0.00304 |
| **Deep RKBS** | **0.00112** |

The difference isn't significant, deep RKBS does slightly outperform deep RKHS in this setting. Since, Deep RKHS is a subset of Deep RKBS we can see the practical significance of this method. Although a more thorough empirical study is required.

For this experiment we have also implemented a small library which contains implementation of multiple RKBS kernels, the library also allows to easily Create deep Kernels by simply specifying the kernel and its ranges.

## 5 RELATED WORKS

While concatenated kernel learning methods such as deepGP have been a well-established line of research, deep RKBS is still in this infency. although there are some very promising recent works in this direction such as Bartolucci et al. (2021), but to the best of authors knowledge, concatenated RKBS kernel learning is not been presented in the literature so far.

## 6 CONCLUSION

We proposed a general framework to construct deep RKBS Kernels, which give rise to a new direction in concatenated kernel learning literature. we started by presenting the general RKBS and its reproducing kernel. we then define a new class of concatenated kernel learning in RKBS spaces and presented a representer theorem along with some examples of deep RKBS, we then derived the connection between the classical concatenated kernel learning methods which includes DKL, MLMKL, DeepGP among other, we finished our paper by describing our new Library to do experiments with Deep RKBS as well as showed some preliminary results of comparing deep RKHS and deep RKBS.

### ACKNOWLEDGMENTS

We would like to thank ICLR committe for organising ICLR CoSubmitting Summer 2022 this year.

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

## A   APPENDIX

You may include other additional sections here.

