# OpenReview forum: "Deep banach space kernels"
_ICLR.cc/2022/Conference — ICLR 2022 Submitted_

### Official Review · Reviewer_5Wv8 · 2021-10-21

**Correctness:** 4
**Technical Novelty And Significance:** 1
**Empirical Novelty And Significance:** 1
**Recommendation:** 3
**Confidence:** 3

**Main Review:**

There isn't a clear division between a background and contributions section, so it is difficult to see the distinction between the background and the contributions.  Nonetheless, it appears that the only contributions are to notice that deep kernel learning can be understood within their framework (Sec. 3.2) and to do a toy experiment on 2D data (Sec. 4).  This is simply not sufficient for a paper at a major machine learning conference, so I am forced to recommend rejection.

**Summary Of The Paper:**

The paper proposes to use kernels from a reproducing kernel Banach space, instead of a reproducing Hilbert space in tasks such as deep kernel learning.

**Summary Of The Review:**

Very limited theoretical and empirical results.

---

### Official Review · Reviewer_kbJr · 2021-11-02

**Correctness:** 4
**Technical Novelty And Significance:** 3
**Empirical Novelty And Significance:** 2
**Recommendation:** 3
**Confidence:** 4

**Main Review:**

While I like the idea underlying this paper I found the paper somewhat unsatisfying.  Specifically:

- unclear/lack of attribution: particularly for the background material, several theorems and definitions are taken almost word-for-word from Lin et al, 2019b.  For example the definition of RKBS corresponds to definition 1.1 in Lin et al, the feature-map construction is the boxed text on page 4 of Lin et al, and theorem 2.1 is the same in both this and Lin et al.  While I understand that re-wording is often tedious, at the very least a simple parenthetical citation (for example "Theorem 2.1 (see theorem 2.1 in Lin et al): ...") seems to be called for.

- Related to the above, theorem 2.2 uses notation $A^\vdash$ that isn't (as far as I can tell) introduced at all in the submission, which makes understanding somewhat difficult.

- Continuing on the question of notation, what is $e_{k_2}$ in section 3.1?  I'm working on the assumption that this is some form of basis vector set (after all this is vector-valued regression) but this needs to be clarified.

- Perhaps I am wrong, but (3.13) looks like it would, in practice, act like an RKBS with a N hyper-parameters and \mathcal{O} (N^2) computation time.  I am curious how this affects the performance of an algorithm using such a kernel - in my experience kernels this complicated can have a severe effect on training and evaluation times, particularly for large datasets!

- The experimental section is unsatisfactory.  A single experiment is described, but the size of the training set is not specified.  50 trials are completed, but no error bars are included.  A link to code is given but the algorithm is not described and training issues (training times etc) are not discussed.

To be clear I think this avenue of research is fascinating and potentially important, but I feel that the deficits of the paper need to be addressed first.

**Summary Of The Paper:**

The paper presents a means of constructing deep (concatenated) reproducing kernel Banach space kernels.  The main contribution appears to be a representor theory for concatenated Banach space kernels.  Some experimental results are provided to accompany this result.

**Summary Of The Review:**

- The premise is interesting.
- The clarity of attribution in the background section needs to be worked on.
- "Niche" notations used but not defined.
- Experimental results on a single dataset without important details (training set size, training times, details of training algorithm etc0.

---

### Official Review · Reviewer_57b6 · 2021-11-04

**Correctness:** 2
**Technical Novelty And Significance:** 1
**Empirical Novelty And Significance:** 1
**Recommendation:** 1
**Confidence:** 5

**Main Review:**

This submission looks more like a working draft rather than a conference paper. In particular,

**There are many typos**, making the reading very difficult (I am highlighting a few of them, only on page 1):
- which *have* their own limitation (abstract)
- that uses *functions* from Reproducing Kernel Banach Spaces (abstract)
- and *test* it on a range on ML tasks (abstract)
- the acronyms MLMKL, DGP, DKL are never introduced (abstract and later)
- *W*e are still (1st paragraph)
- *kernel from Hilbert Space* makes little sense (1st paragraph)
- There has *been* (1st paragraph)
- *W*e then examine (2nd paragraph)
- *two layer RKBS kernel* makes little sense (2nd paragraph)
- the connect*ion* with multiple (2nd paragraph)
- *W*e also talk about *the* library (2nd paragraph)
- the bibliography entry [Scholkopf and Smola] seems wrong

**The maths are not rigorous**, to the extent they are almost incorrect:
- the introduction of RKBSs is very confusing, as the proposed definition looks more like the RKHS definition to me, see e.g., Definition 4.18 in "Support Vector Machines" by Steinwart and Christmann, 2008
- Theorems 2.1 and 2.2, that are not part of the contribution, should be explicitly linked to the paper they come from
- undefined notation in eq. (2.9)
- I would like to point out the paper "Autoencoding any Data through Kernel Autoencoders" by Laforgue et al., 2019 which considers a similar framework as the one introduced at the top of page 3. A Representer Theorem is also proved therein, as in [Bohn et al. 2019], except that it applies to potentially infinite dimensional output Hilbert spaces
- the introduction of the $\Theta_l$ is not understandable
- In section 3.2 the proof is said to be given in the Appendix, but the latter is empty

**The contribution is very limited**:
- the Representer Theorem (RT) being mainly due to orthogonality properties, it is not surprising to recover it for a class of functions that share this property with RKHSs
- I cannot see any novel idea in the proofs, so proving another RT seems not a sufficient contribution for acceptance
- I cannot find any motivation for the presented results
- the authors themselves acknowledge that their experimental contribution is very limited

**Summary Of The Paper:**

This paper proves a Representer Theorem for the composition of functions from Reproducing Kernel Banach Spaces (RKBSs).

**Summary Of The Review:**

This submission looks more like a working draft rather than a conference paper.

---

### Decision · Program_Chairs · 2022-01-20

**Decision:**

Reject

**Comment:**

The paper develops kernel functions in Banach spaces. However the results seem to be preliminary and further development is needed before
the manuscript can be published. Reviewers point out several errors and also author/authors have graciuously agree with the suggestion
that they will incorporate all the feedback in future submissions.